# Effect of Stress Path on the Failure Envelope of Intact Crystalline Rock at Low Confining Stress

**Shantanu Patel** [1,*] and **C. Derek Martin** [2]

1   Department of Mining Engineering, Indian Institute of Technology Kharagpur, Kharagpur WB 721302, India
2   Department of Civil and Environmental Engineering, University of Alberta,
    Edmonton, AB T6G 2R3, Canada; derek.martin@ualberta.ca
*   Correspondence: shantanu.patel@mining.iitkgp.ac.in

**Abstract:** Numerical modelling is playing an increasing role in the interpretation of geological observations. A similar phenomenon is occurring with respect to the interpretation of the stress–strain response of intact rock measured in laboratory tests. In this research, the three-dimensional (3D) bonded particle model (BPM) with flat-jointed (FJ) contact was used to investigate the impact of stress paths on rock failure. The modified FJ contact model used for these studies numerically captured most of the intact rock behavior of Lac du Bonnet granite observed in the laboratory. A numerical simulation was used to track the behavior of this rock for different stress paths, starting with uniaxial tension and compression loading conditions. The migration from uniaxial tension to triaxial compression is challenging to simulate in physical laboratory tests but commonly observed around underground excavations. The numerical modelling methodology developed for this research tracks this stress path and the impact of the intermediate stress on peak strength at low confinements, commonly found around underground excavations.

**Keywords:** stress path; confined extension test; flat-jointed bonded particle model; intermediate principal stress

## 1. Introduction

Design parameters of rock typically start with the results from laboratory testing carried out for the stress conditions and the stress path expected in the field; however, as noted by Brady and Brown [1], this can rarely be achieved. Near the boundary of an underground opening, the rock can be subjected to a variety of stress conditions ranging from confined extension to confined compression [2,3]. Kaiser et al. [4] used in situ strain cells to track the stress path followed by the excavation boundary. While the rock near the boundary was initially subjected to all-around compressive stresses, by the end of the excavation those stresses had converted to confined extension. The microscale stress path followed and its damage on an intact rock during coring was investigated numerically by Bahrani et al. [5]. Developing a laboratory test methodology that can track the stress path reported by Kaiser et al. [4] is not yet possible. Consequently, the failure of rock is typically tested using simplified stress paths, such as direct tension, uniaxial compression, and triaxial compression. A consequence of this simplification is the limited ability to track the progressive fracture of rock from confined compression to confined extension.

Analyses of stress-induced failures observed around underground excavations show that the mobilized strength of sparsely fractured and massive rock masses is typically one-third to one-half the strength measured in laboratory compression tests [6–8]. The difference between the monotonic stress path used in laboratory strength tests vs. the load–unload complex stress path followed by the rock

mass around the tunnel is often suggested as the primary reason for this difference [4,9–13]. Advances in numerical modelling provide an opportunity to revisit this assumption.

In this research, the three-dimensional (3D) BPM with FJ contact [14] was used to investigate the impact of stress paths on rock failure. Using the FJ contact model, Refs [15,16] show that most of the intact rock behavior of Lac du Bonnet granite observed in the laboratory can be captured numerically. A numerical simulation that followed the laboratory behavior of Lac du Bonnet granite was used to track the behavior of this rock under confined extension and triaxial loading conditions for different stress paths. The impact of the intermediate stress on peak strength at low confinements, commonly found around underground excavations, was also investigated.

## 2. Background

### 2.1. Stress Paths Observed in the Field and in Typical Low Confinement Laboratory Tests

The stress paths taken by laboratory sample tests in direct tension [17], uniaxial compressive strength (UCS) [18], and triaxial tests are shown in the $\sigma_1$–$\sigma_3$ plot in Figure 1. In the direct tension and UCS tests, the principal stress monotonically increases from an initial state with zero stress until the sample fails. Similarly, in the triaxial test, a sample is initially taken to a hydrostatic condition and then $\sigma_1$ is monotonically increased until failure occurs.

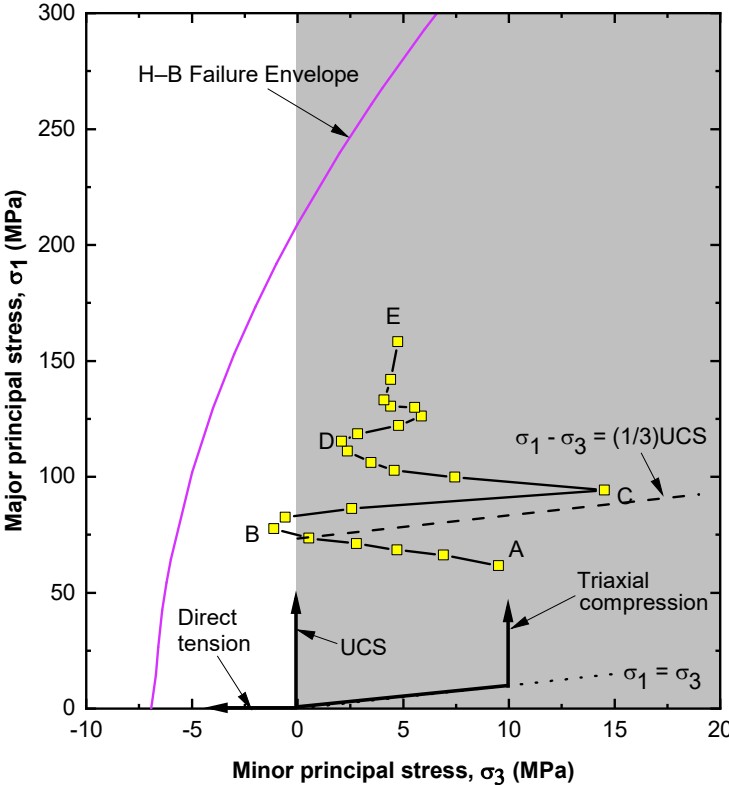

**Figure 1.** Stress path obtained for a point 50 mm above the crown of a mine-by tunnel using 3D numerical modelling [9]. The Hoek–Brown (H–B) failure envelope and the stress path for direct tension, UCS, and triaxial tests are also shown. The thick dashed line represents the stress level above which crack initiates in rock.

The stress path for the rock surrounding an opening can be obtained from direct field measurements during construction (Figure 2) or using 3D numerical modelling of the opening considering the geometry of the opening and the excavation sequence (Figure 1). As shown in Figure 1, the stress path for rock that is 50 mm above the crown is complex. Figure 1 shows the results of numerical modelling of a

mine-by tunnel [9] excavated in Lac du Bonnet granite and the Hoek–Brown (H–B) failure envelope obtained from laboratory triaxial testing of the rock. Point A in the $\sigma_1$–$\sigma_3$ plot in Figure 1 represents the in situ stress condition. Path ABCDE shows the change in $\sigma_1$ and $\sigma_3$ values as construction progresses towards point A and passes through it. The dashed line near point B, $\sigma_1 - \sigma_2 = (1/3)$UCS, represents the stress level around which crack initiates in the rock. When the stress path approaches point B, microcracks start forming in the rock and, thereafter, the damage process leading to failure depends upon the stress path it takes.

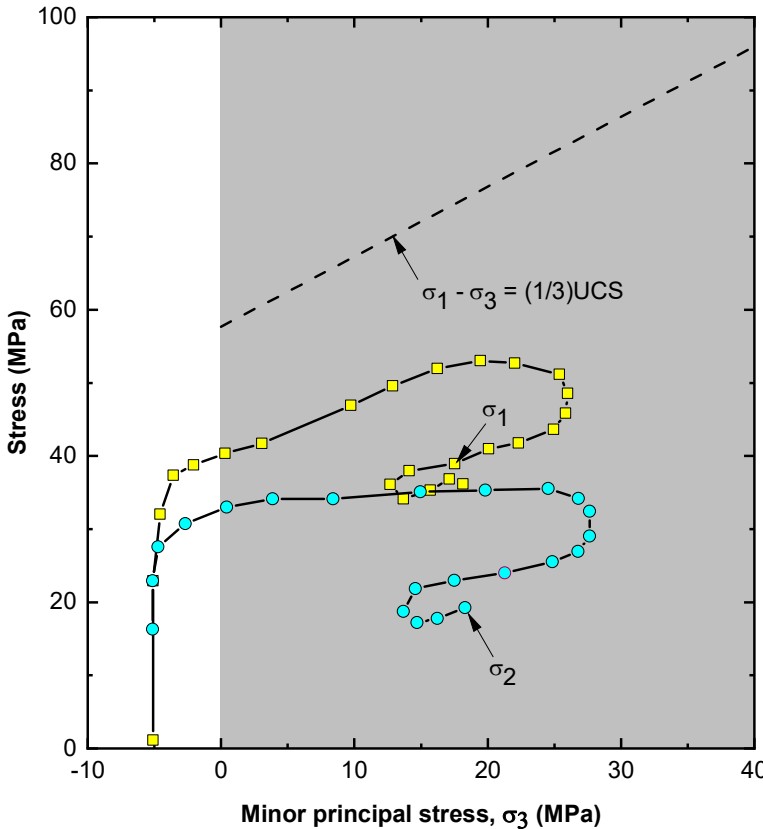

**Figure 2.** Stress path obtained from field measurements using CSIRO HI stress cells for $\sigma_1$ and $\sigma_2$ versus $\sigma_3$ in the hanging wall of 565#6 stope, Winston Lake Mine [4]. Note: the rock in situ stress changes from initially confined compression to confined extension.

Kaiser et al. [4] used CSIRO HI stress cells for stress measurements to obtain the stress path directly from field measurements in the hanging wall of the a mine stope. Figure 2 shows the stress path taken by $\sigma_1$ and $\sigma_2$ versus $\sigma_3$. Comparing the initial and final stress states shows the presence of a stress rotation in addition to changes in the magnitudes of the principal stresses. Figures 1 and 2 show that, during the stress path, rock can come across a condition where one of the principal stresses can be compressive and the other tensile. This is called a confined extension condition and is explored in the next section.

*2.2. Laboratory-Confined Extension Test on Rock*

Confined tensile fracturing occurs in rock under mixed tensile and compressive stress conditions, and a test for this situation was proposed by Brace [19]. As shown in Figure 3, all-around compressive stress (*P*) and vertical compressive force (*F*) are initially applied to the model. The load (*F*) on the top and bottom platens is then gradually reduced until the specimen fractures. The confining stress in the

curved part of the sample generates the tensile stress in the central part. The compressive and tensile stresses at failure are then calculated from Equations (1) and (2):

$$\sigma_1 = \sigma_2 = P, \tag{1}$$

$$C = \sigma_3 = -\frac{P'(A_h - A_t)}{A_t} + \frac{F}{A_t}, \tag{2}$$

where $C$ is the axial stress (positive in tension), $F$ is the axial force (negative $F$ acts towards sample), $P$ is the confining pressure (positive), $A_h$ is the head area, and $A_t$ is the throat area. The confined extension test conducted is called the triaxial confined extension test when $\sigma_2 = \sigma_1$. Rock tested in triaxial confined extension and triaxial tests in the same range of confinement shows that samples tested in triaxial confined extension fail at a considerably higher peak stress. The data available in the literature for confined extension are very limited. Two rocks subjected to extensive triaxial confined extension tests are Carrara marble [20] and Berea sandstone [21], shown in Figure 4a,b. Patel and Martin [22] compared the biaxial strength ($\sigma_1 = \sigma_2 =$ compressive, $\sigma_3 = 0$) obtained from laboratory UCS testing of these two rocks and observed a clear impact of $\sigma_2$ on peak strength. If the confined extension test results ($\sigma_1 = \sigma_2 > \sigma_3$) are compared with triaxial test results ($\sigma_1 > \sigma_2 = \sigma_3$), there is a clear mismatch for both rock types (Figure 4). In the UCS tests, the difference is 41% and 22% for Carrara marble and Berea sandstone, respectively. The reason for this is yet unknown and could be due to the different stress path taken or the impact of $\sigma_2$. This is explored in the next sections.

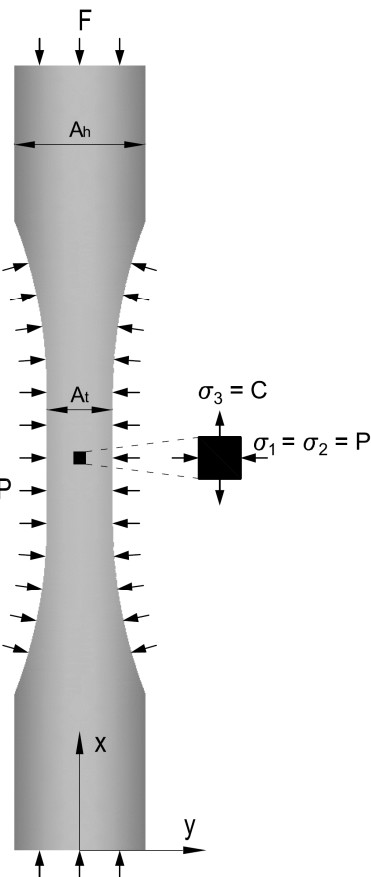

**Figure 3.** Confined extension test on a dog-bone-shaped rock sample (after Brace, 1964).

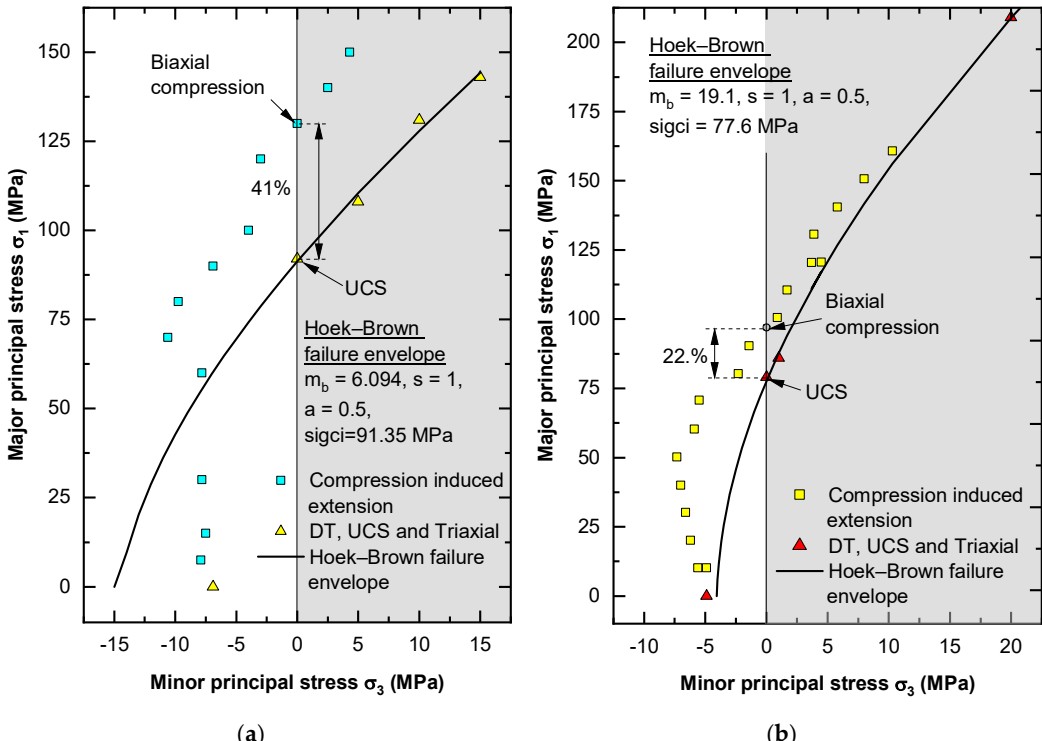

**Figure 4.** Results of confined extension test for (**a**) Carrara marble [23] and (**b**) Berea sandstone [21]. Direct tension (DT), UCS, triaxial compression, and confined extension test results as well as the Hoek–Brown failure envelopes are shown. The biaxial strength of the Berea sandstone was calculated using interpolation.

Patel and Martin [22] used the flattened Brazilian test to test the sample in confined extension with $\sigma_2 = 0$, where increased confinement was tested based on samples with an increased depth of flattening. However, the maximum confinement achieved using the flattened Brazilian test was 37% of the UCS. The FJ BPM was therefore used to investigate confined extension.

### 2.3. Flat-Jointed Bonded Particle Model

In the BPM, the mineral grains in a rock sample are represented by rigid spheres bonded at their contacts. This model has many applications from fundamental research on rock mechanics [24] to large-scale applications [25]. Potyondy and Cundall [26] first introduced the parallel BPM, which had a few shortcomings. For example, it cannot capture the high value of the UCS/$\sigma_t$ ratio of low porosity rocks [7] and requires cluster logic to increase the friction angles to more realistic values and increase the triaxial strength after calibrating the UCS [26]. Therefore, Potyondy [14] suggests the FJ BPM to address these limitations.

Figure 5a shows the 3D FJ model used in this investigation. The mineral grains are represented by grey spheres and the contacts by disk-shaped blue FJs. The FJ contacts are further discretized along the radial and tangential directions as shown in Figure 5b. Each element in the FJ contact can fail separately when an external load is applied. Potyondy [14] shows that the moment-carrying capacity of partially failed FJ is essential to produce the high UCS/$\sigma_t$ ratio. Patel and Martin [16] further investigated the role of initial microcracks present in a rock sample. They showed that an FJ sample with initial microcracks is essential to capture the initial nonlinearity, crack initiation stress, and tensile and compressive Young's moduli in addition to the UCS/$\sigma_t$ ratio of intact rock.

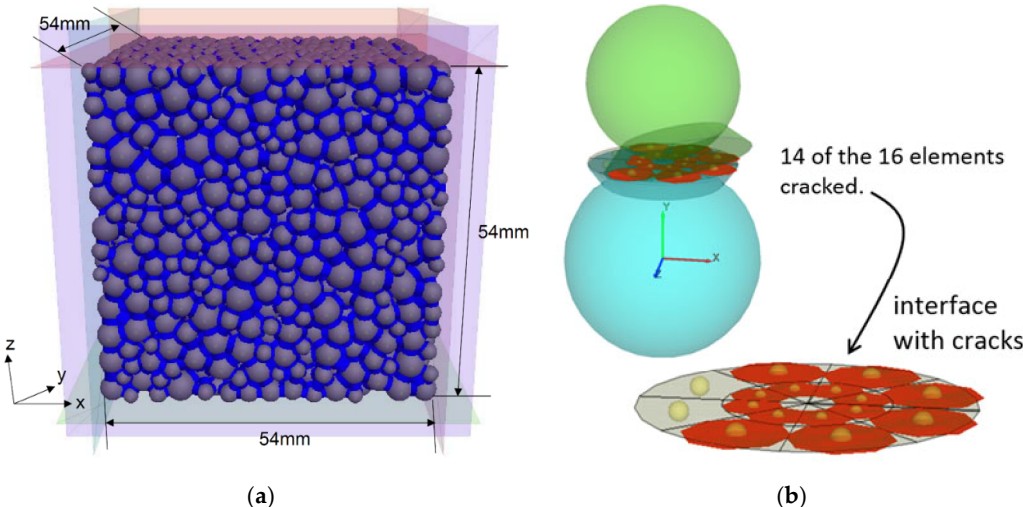

|                      (**a**)                      |                      (**b**)                      |

**Figure 5.** (**a**) The cubical FJ BPM made up of 3618 grains and 17,110 FJ contacts used in the investigation and (**b**) representation of a partially fractured FJ [24]. In the fractured FJ, 14 of the 16 elements are cracked due to bending.

## 3. Calibration of Microparameters for Lac du Bonnet Granite for the BPM

The first step in the FJ BPM is to obtain the microparameters for the grains and FJ contacts. These microparameters cannot be obtained experimentally and an iterative method is needed. Table 1 shows that an FJ contact requires 18 parameters for calibration, along with the parameters associated with grain-size distribution and linear material group. These parameters are described in the material modelling support manual [27]. Patel [28] describes how a set of microparameters can be efficiently obtained for a numerical rock sample to consistently reproduce the results of a real rock sample. The FJ model was used in this study to capture the rock behavior observed in the laboratory during UCS, triaxial, and direct tension tests. Using a single mineral grain type and an average of 15 particles along the width of the sample, it was possible to produce reasonable (a) bi-modularity of the rock, (b) initial nonlinearity in the axial stress–strain curve during UCS and triaxial tests, (c) crack initiation stress and crack damage stress in an UCS test, (d) ratio between the UCS and direct tension test, and (e) other responses of the laboratory sample during triaxial tests. The final set of parameters used in this investigation, which mimic the behavior of Lac du Bonnet granite, is presented in Table 1.

**Table 1.** List of microparameters obtained after calibration for the numerical rock sample. Parameters required for grain and material genesis are also listed [27].

| Parameter | Value |
|---|---|
| *Associated with grain size distribution:* | |
| Minimum grain diameter | 2.2 mm |
| Grain-size ratio | 2.3 |
| *Associated with material genesis:* | |
| Width of sample | 54 mm |
| Height–width ratio | 1 |
| *Associated with FJ material group:* | |
| Installation gap | 1.31 mm |
| Bonded fraction | 0.65 |
| Gapped fraction | 0.35 |
| Slit fraction, derived | 0 |
| Initial surface-gap distribution, mean | 0.002 mm |
| Initial surface-gap distribution, standard deviation | 0 |
| Elements in radial direction | 1 |

**Table 1.** *Cont.*

| Parameter | Value |
|---|---|
| Elements in circumferential direction | 3 |
| Radius-multiplier code | 0 |
| Radius-multiplier value | 0.577 |
| Effective modulus | 135.8 GPa |
| Stiffness ratio | 1.2 |
| Friction coefficient | 1.4 |
| Tensile-strength distribution, mean | 41.6 MPa |
| Tensile-strength distribution, standard deviation | 0 |
| Cohesion distribution, mean | 203 MPa |
| Cohesion distribution, standard deviation | 0 |
| Friction angle | 43.2° |
| *Associated with the linear material group:* | |
| Effective modulus | 135.8 GPa |
| Stiffness ratio | 1.2 |
| Friction coefficient | 2.2 mm |

### 3.1. Intergranular Stiffness Ratio and the Formation of Microcracks in the Sample

The ratio between the normal and shear stiffness (krat) between the mineral grains plays an important role in microcrack formation in the sample during UCS testing. Keeping the normal stiffness constant but increasing the ratio allows the intergranular slide to happen more easily, forming an early generation of crack and a higher Poisson's ratio for the numerical sample. To match the Poisson's ratio of the material, a common practice is to assume a ratio value of around 1.5 (e.g., [15]). However, we found out that, in the FJ sample during the UCS test, tensile cracks start forming at an early stage of loading (around 10% of peak) and the curve between the total number of the cracks in the numerical sample versus the axial load does not support laboratory observations. This is shown in Figure 6, where the crack generated for the numerical sample with krat = 1.6 calibrated by the laboratory macroparameters is compared with the laboratory Lac du Bonnet granite samples.

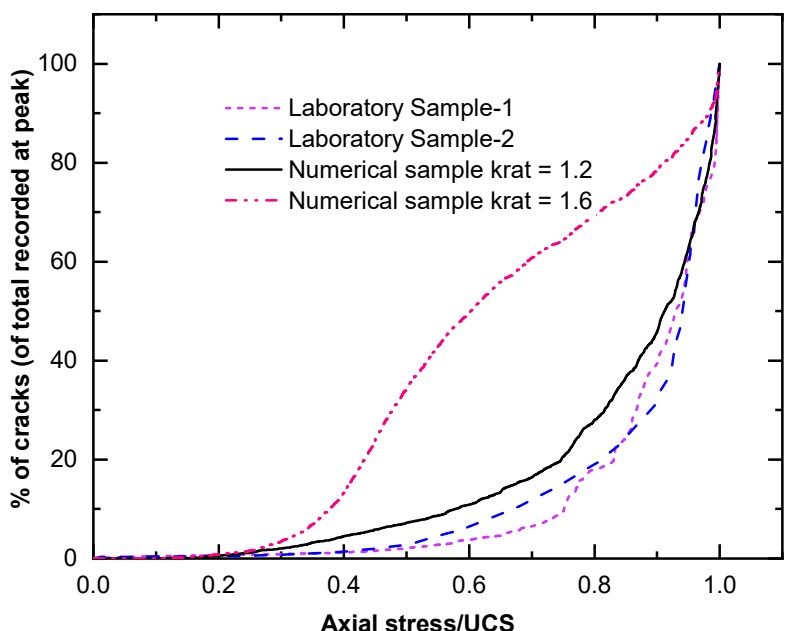

**Figure 6.** Comparison of microcrack formation in the numerical sample with krat = 1.6 or 1.2 and the laboratory Lac du Bonnet granite sample. The sample-1 result is from Martin [29] and the sample-2 result is from Eberhardt et al. [30]. Note the numerical samples with krat = 1.6 or 1.2 are calibrated by the same macroparameters as those for Lac du Bonnet granite.

When numerical models were studied in both compression and tension, the early generation of cracks had a minor impact on the UCS test but a major impact on the confined extension tests because of the stress path. When a sample subjected to compression was tested in tension, the early microcracks considerably reduced the strength of the rock. To reduce the formation of early cracks and match the laboratory crack formation for Lac du Bonnet granite, the krat was reduced to 1.2 keeping all other microparameters constant. This changed most of the macroparameters obtained for the numerical sample. The model was then recalibrated to the direct tensile strength, Young's moduli, crack initiation stress, and peak strength from the UCS test. The corresponding set of microparameters is given in Table 1. Figure 6 shows the crack formation in the calibrated numerical sample vs. the laboratory observations. A krat of 1.2 considerably reduces the crack number and more closely matches the crack formation noted in the laboratory. Table 2 also shows that the data match most of the microparameters observed in the laboratory.

**Table 2.** Comparison of macroresponses observed in the laboratory Lac du Bonnet granite and numerical samples.

| Case | $E_c$ (GPa) | $\nu_c$ | $\sigma_{ci}$ (MPa) | $\sigma_{cd}$ (MPa) | UCS (MPa) | $E_t$ (GPa) | $\sigma_t$ (MPa) | $E_t/E_c$ |
|------|------|------|------|------|------|------|------|------|
| Lab | 70.5 | 0.26 | 88.6 | 163.3 | 221.7 | 45.8 | 10.6 | 0.65 |
| Numerical | 69.3 | 0.1 | 97.7 | 208.1 | 219.6 | 47.7 | 10.8 | 0.69 |

*3.2. Comparison of Laboratory Response of the Lac du Bonnet Granite Sample with the FJ Numerical Sample for the UCS Test*

The stress–strain curve obtained for a numerical sample with tensile and shear cracks generated due to an increase in axial strain is shown in Figure 7. The initial nonlinearity in the stress–strain curve, which is typical of UCS and triaxial rock testing, was captured by considering microgaps between the mineral grains present in the laboratory sample. The numerical sample exhibited bi-modularity (different $E_t$ and $E_c$) when stress-released microfractures were included in the model. The ratio between the simulated $E_t$ and $E_c$ was 0.69, which is within 5% of that obtained from laboratory testing. The crack initiation stress ($\sigma_{ci}$) is the point in the axial stress vs. radial strain curve where the curve deviates from linearity after the initial elastic region; this value was 97.7 MPa for the numerical sample vs. 88.6 MPa in the laboratory. When the $\sigma_{ci}$ point was projected down in Figure 7, it matched the point where tensile microcracks started forming rapidly in the sample. Because only tensile cracks formed before $\sigma_{cd}$, these cracks cannot be reflected in the axial stress versus axial strain curve. As shown in Figure 7, the axial stress versus axial strain curve deviates from linearity when the shear cracks start in the sample. These observations are in line with the laboratory findings of many researchers, such as Brace et al. [31] and Bieniawski [32]. The macroparameters for the numerical sample are compared with the Lac du Bonnet granite sample in Table 2.

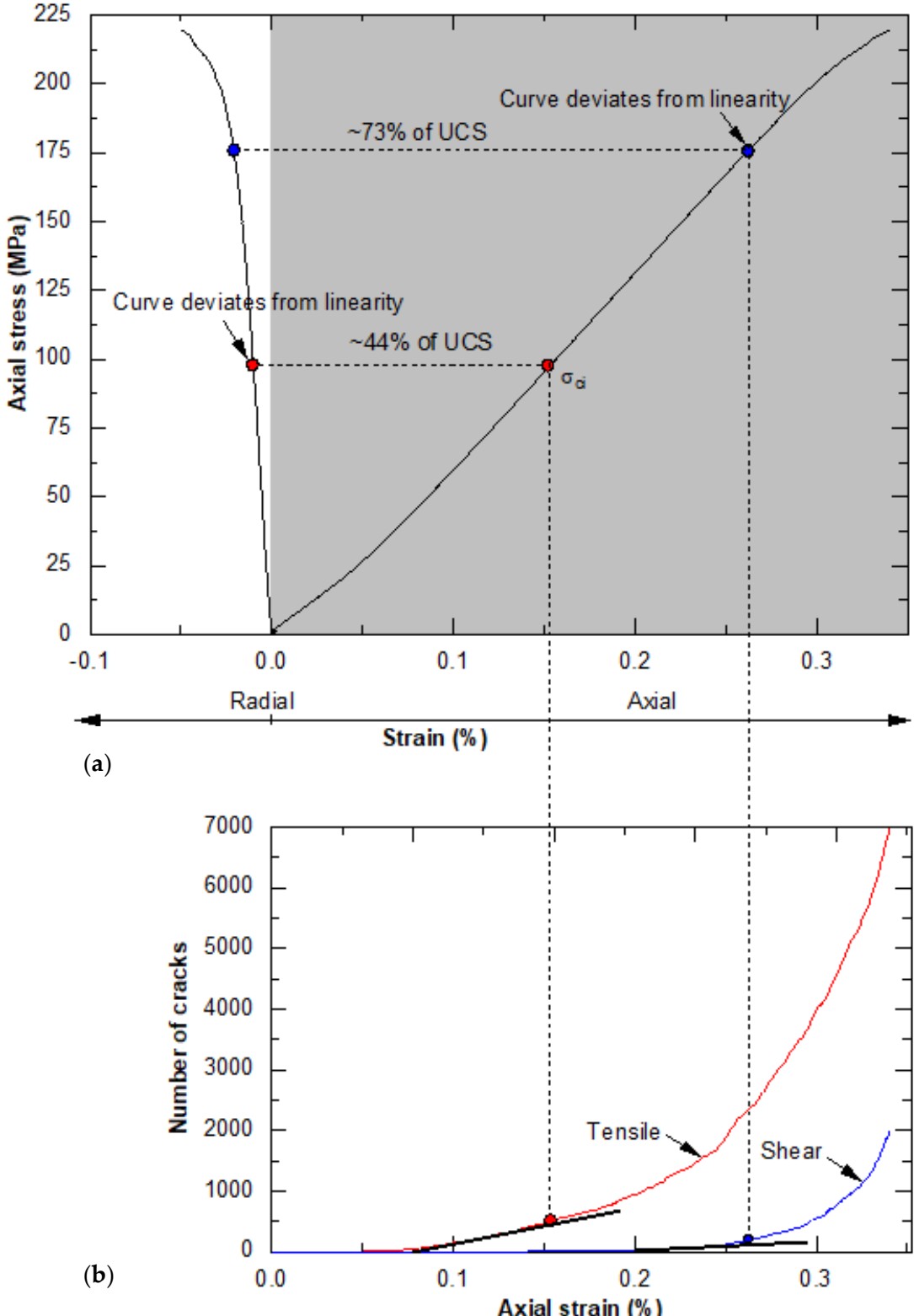

**Figure 7.** Results of the UCS test for the numerical sample. (**a**) shows the axial stress versus the axial and radial strain and (**b**) shows the generation of tensile and shear cracks with axial strain.

## 4. Confined Extension Test Using the FJ BPM

As shown in Figures 1 and 2, rock around an opening may encounter confined extension conditions when $\sigma_1$ is compressive and $\sigma_3$ is tensile. Schöpfer et al. [33] and Huang and Ma [34] use the parallel BPM to investigate rock under confined extension. The present study used the FJ code to investigate the intact rock behavior in confined extensions after obtaining acceptable results for the numerical sample compared to the typical laboratory test samples.

The central part of the sample, Figure 3, used in the laboratory testing of rock in confined extension passes through the stress path ODEC, as shown in Figure 8. The sample at zero confinement (point O) is subjected to hydrostatic stress (point D). The $\sigma_3$ is then gradually reduced, keeping $\sigma_1$ and $\sigma_2$ constant until the sample fractures at point C. Samples that fail along such a stress path are influenced by $\sigma_2$ [35]. For the investigation of the confined extension on the numerical sample (Figure 3) with no $\sigma_2$, the following steps were taken: (A) the cubical sample shown in Figure 5 was hydrostatically confined to a particular magnitude of $\sigma_1$ (point D, Figure 8); (B) keeping $\sigma_1$ constant, $\sigma_2$ and $\sigma_3$ were simultaneously reduced (point E, Figure 8); (C) the top and bottom 5 mm of the sample were then held and pulled in opposite directions keeping $\sigma_1$ constant and $\sigma_2$ at zero until the sample failed at point C (Figure 8). The x-coordinate gives the magnitude of $\sigma_3$ for the corresponding $\sigma_1$; and (D) steps A–C were then repeated for different magnitudes of $\sigma_1$ to obtain different failure points in confined extension.

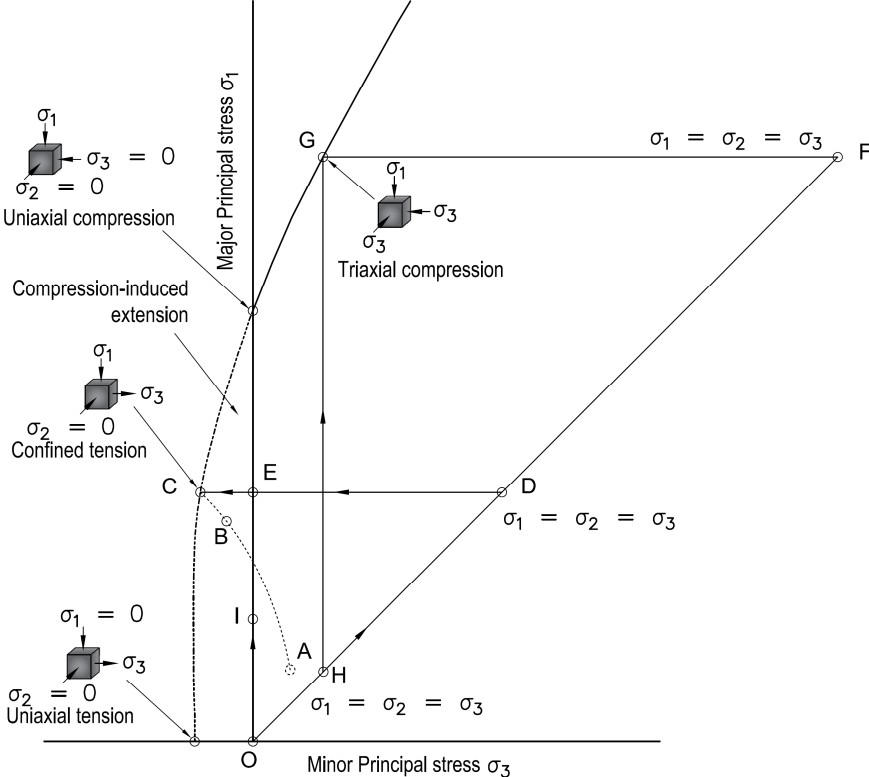

**Figure 8.** A typical failure envelope for rock showing the stress path during probable field-confined extension (ABC, unloading), confined extension (ODEC and OFG), and triaxial (OHG) tests. Note: the location of point A depends upon the boundary condition to which the rock is subjected in the field.

## 5. Results of FJ Modelling

### 5.1. Influence of Stress Path on a Confined Extension Test

As shown in Figure 8, point A represents an in situ stress condition with $\sigma_1 \geq \sigma_2 \geq \sigma_3$. Due to the excavation, i.e., change in the boundary condition, the rock at this point may fail in confined extension following path ABC. However, this stress path is case dependent and can be complex. The path followed by the conventional confined extension test using the dog-bone-shaped geometry is ADEC (Figure 8). The microfracture formation in rock and its peak strength are path dependent. To compare the impact of the stress path followed, paths ODEC and OEC were compared for the numerical FJ sample. As shown in Figure 9, more cracks form (533 vs. 470) in the sample when it follows stress path OEC (a low confinement stress path compared to ODEC), although the final stress conditions are similar (90 MPa at E). This influences the peak strength obtained from the samples due to the application of tensile load. As shown in Figure 10, stress paths ODEC and OEC have a 3.6% difference in peak strength.

Figure 11a,b compares the tensile and shear cracks generated in the sample when it moves from point E to point C. The number of shear cracks formed in the numerical sample is greater when the stress path followed in the laboratory is considered, which supports the results of Ramsey and Chester [20]. Stress path ODEC was considered to compare the numerical results with the laboratory results.

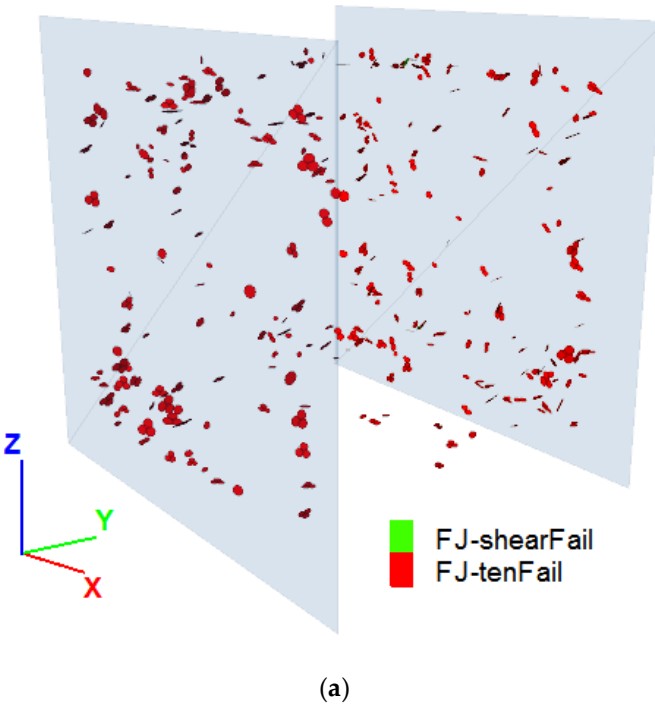

(**a**)

**Figure 9.** *Cont.*

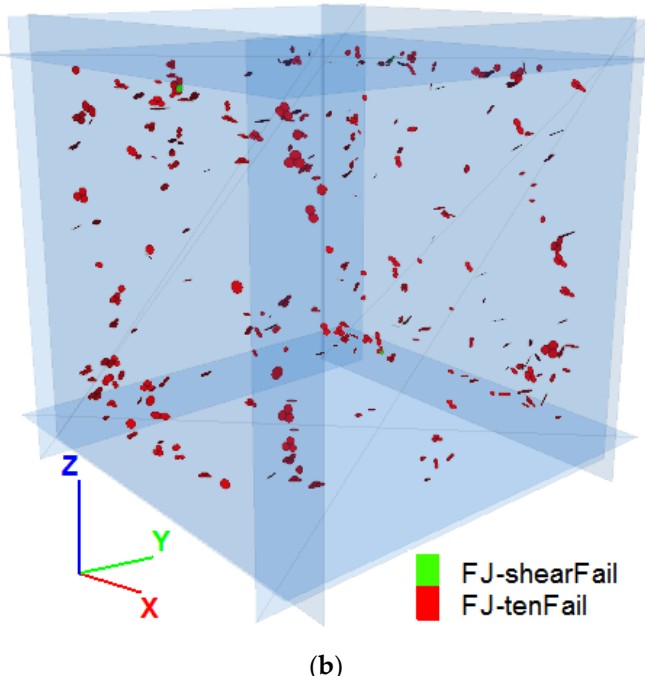

(**b**)

**Figure 9.** Cracks formed in the sample at point E when the sample follows stress path (**a**) OEC, $\sigma_1$ = 90 MPa, $\sigma_2$ = 0 and (**b**) ODEC, $\sigma_1$ = $\sigma_2$ = 90 MPa. (**a**) 533 tensile and 3 shear, (**b**) 470 tensile and 5 shear.

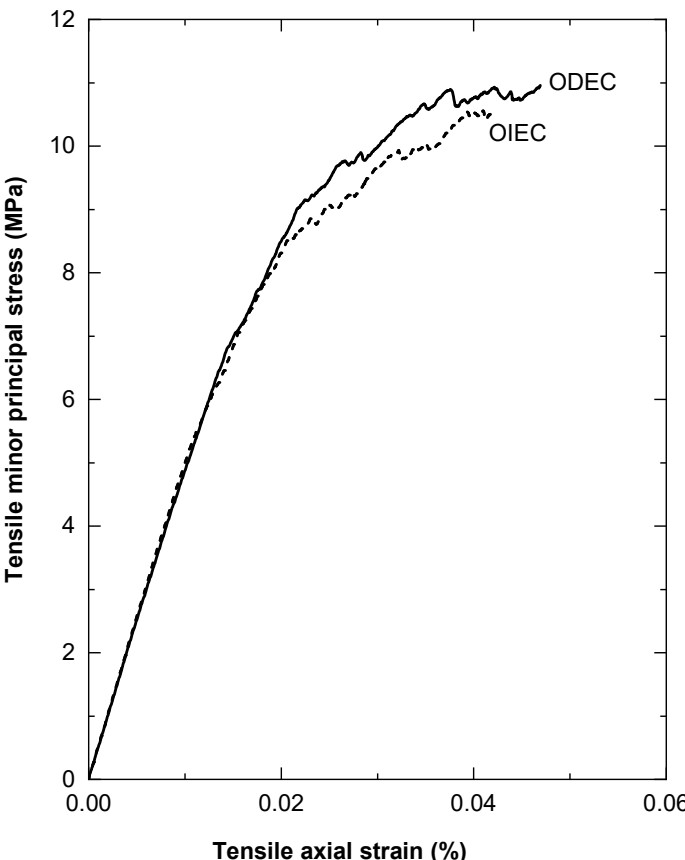

**Figure 10.** Tensile axial stress–strain curve obtained due to the application of tensile stress for stress paths ODEC and OEC (from point E to point C, Figure 8) at $\sigma_1$ = 90 MPa.

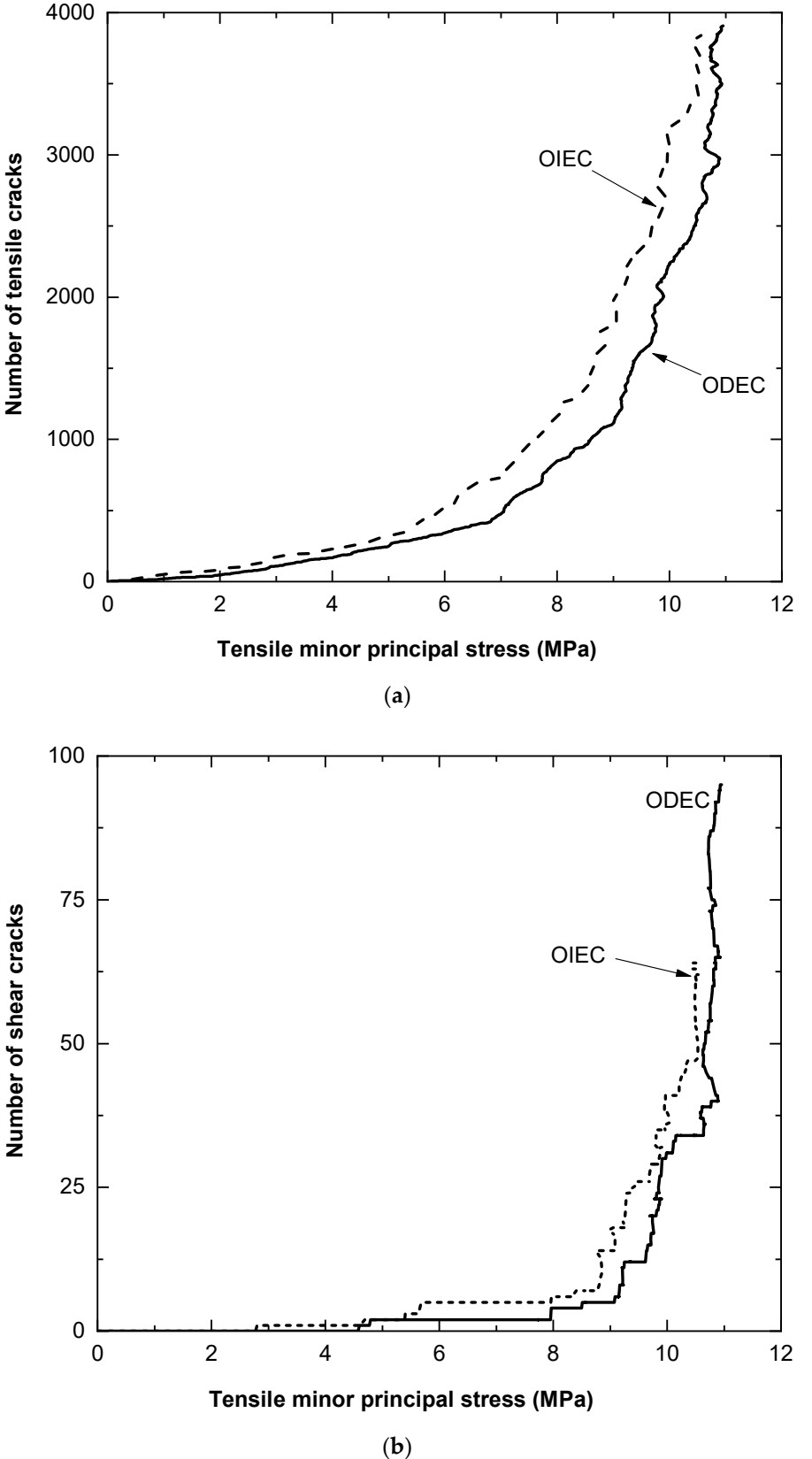

**Figure 11.** Comparison of crack development in the sample with an increase in the tensile strain from point E to point C for stress paths ODEC and OEC: (**a**) tensile cracks and (**b**) shear cracks.

### 5.2. Results of the Confined Extension Test with $\sigma_2 = 0$

Using the steps described in Section 4, confined extension tests were carried out at different magnitudes of $\sigma_1$. The stress–strain results for the tests between points E and C (Figure 8) at three stress levels—low (10 MPa), medium (90 MPa), and high (150 MPa)—are shown in Figure 12. At low confinement, the sample fails similar to a direct tension test sample. However, because of the confinement, the sample fails at a higher tensile stress (i.e., similar to the Brazilian tensile strength being greater than the direct tensile strength for Lac du Bonnet granite). With an increase in confinement, the peak strength of the sample increases (up to around 50% of the UCS value). As shown in Figures 13 and 14, tensile cracking dominates at low confinement, while the percentage of shear cracks increases with increasing confinement. This finding is in agreement with observations by Brace [19] and Ramsey and Chester [20] using confined extension tests on dog-bone-shaped samples.

The confinement in the numerical sample was then increased from 10 MPa to around 80% of the UCS, which is around the $\sigma_{cd}$ of Lac du Bonnet granite. When a sample with confinement more than the crack damage stress is relaxed from point D to point E (Figure 8), equilibrium is not reached and the sample fails. This observation indicates the peak strength obtained after the crack damage stress depends upon the applied boundary condition.

The sample was then further confined beyond the UCS (point F, Figure 8) and $\sigma_2$ and $\sigma_3$ gradually relaxed until the sample failed at point G (Figure 8). The results are plotted and a discussion is presented in Section 6. These results are also compared with the conventional triaxial stress (stress path OFG vs. OHG) in Section 6.

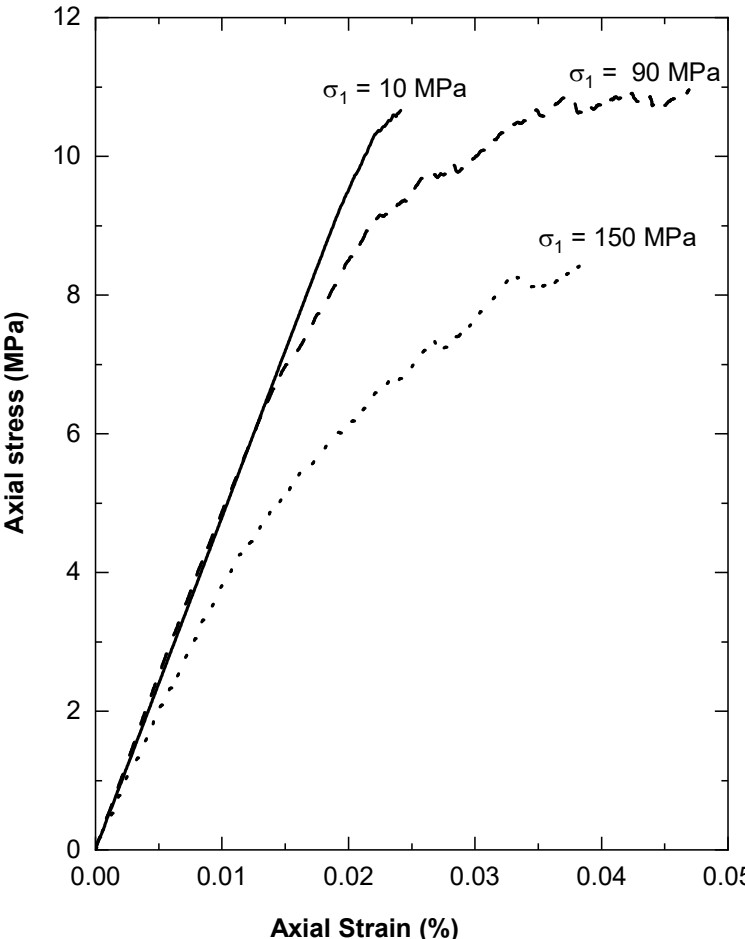

**Figure 12.** Stress–strain curves obtained for numerical confined extension tests at low (10 MPa), medium (90 MPa), and high (150 MPa) confinements (from point E to point C; Figure 8). In all cases, $\sigma_2 = 0$.

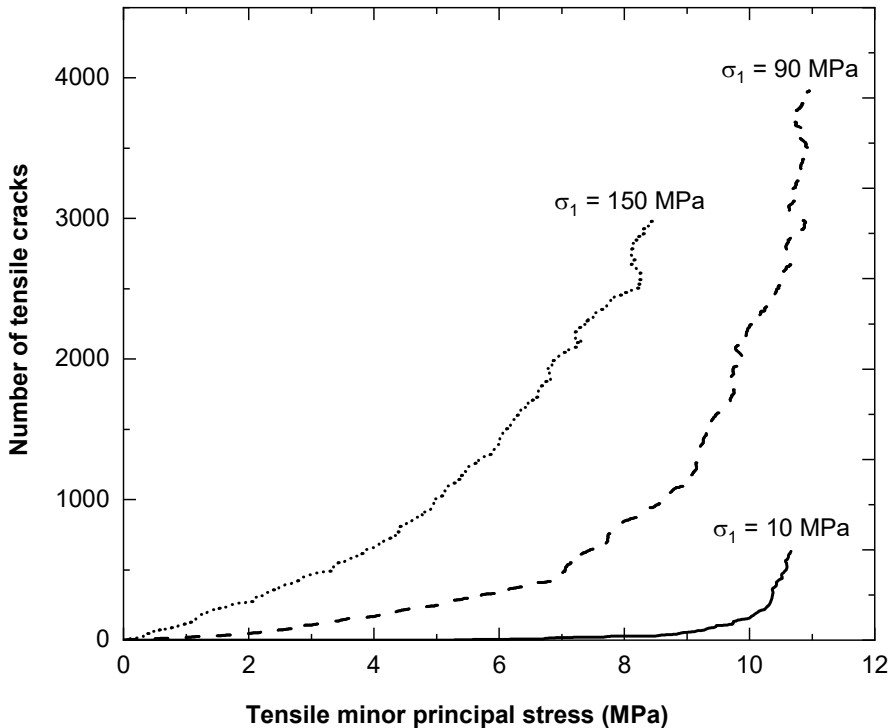

**Figure 13.** Number of tensile cracks formed with an increase in σ₃ for numerical confined extension tests at low (10 MPa), medium (90 MPa), and high (150 MPa) confinements (from point E to point C; Figure 8).

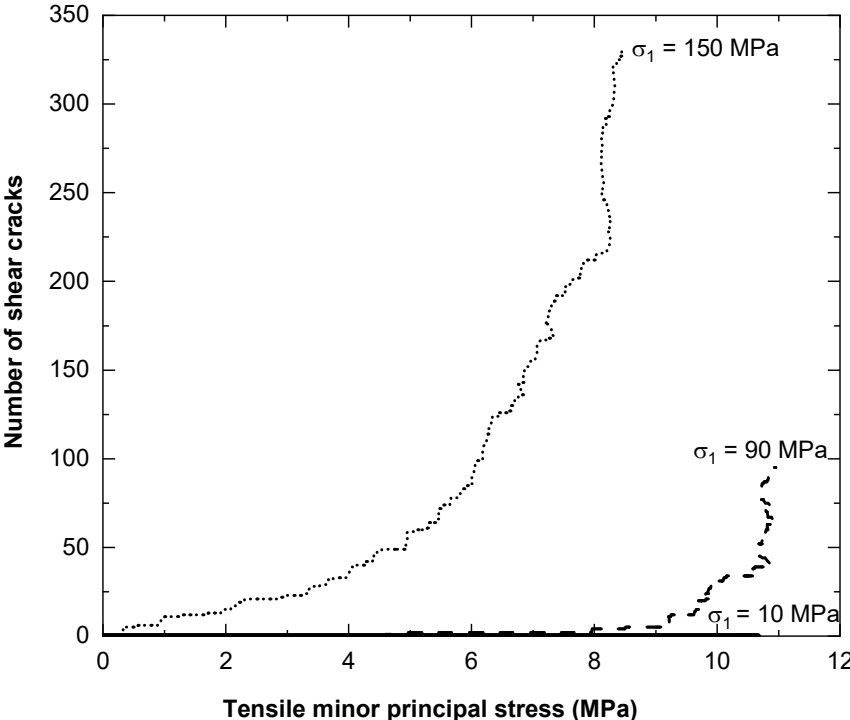

**Figure 14.** Number of shear cracks formed with an increase in σ₃ for numerical confined extension tests at low (10 MPa), medium (90 MPa), and high (150 MPa) confinements (from point E to point C; Figure 8).

### 5.3. Impact of $\sigma_2$ on the Confined Extension Test

One of the main objectives of this study was to examine the influence of $\sigma_2$ on the confined extension test of rock. A confined extension test on a numerical sample was conducted by loading the sample hydrostatically to point D (Figure 8). After point D, $\sigma_3$ was gradually reduced while keeping $\sigma_1$ and $\sigma_2$ constant. The sample was then extended in the axial direction to fail the sample in confined extension. Figure 15 compares the formation of cracks in the sample after $\sigma_3$ is unloaded (point E; Figure 8) and the sample fails in extension (point C; Figure 8) at a $\sigma_1$ of 90 MPa and three intermediate stress levels (90, 45, and 0 MPa). The cracks formed at point E (Figure 8) in the samples are random. The number of cracks formed is less when additional confinement is provided by $\sigma_2$. $\sigma_2$ also prevents the formation of tensile cracks in the model. The shear crack percentage in the case where $\sigma_2$ is equal to $\sigma_1$ is greater than when $\sigma_2 = 0$. As shown in Figure 16, the peak strength increases from 11.0 MPa at $\sigma_2 = 0$ to 19.2 MPa when $\sigma_2$ is kept the same as $\sigma_1$. Figure 17 compares the crack formation from points E to F (with an increase in extensile stress in the numerical sample). The number of shear cracks formed in the sample is greater when $\sigma_2$ is applied to the sample. This is in agreement with confined extension results on dog-bone-shaped Carrara marble [20].

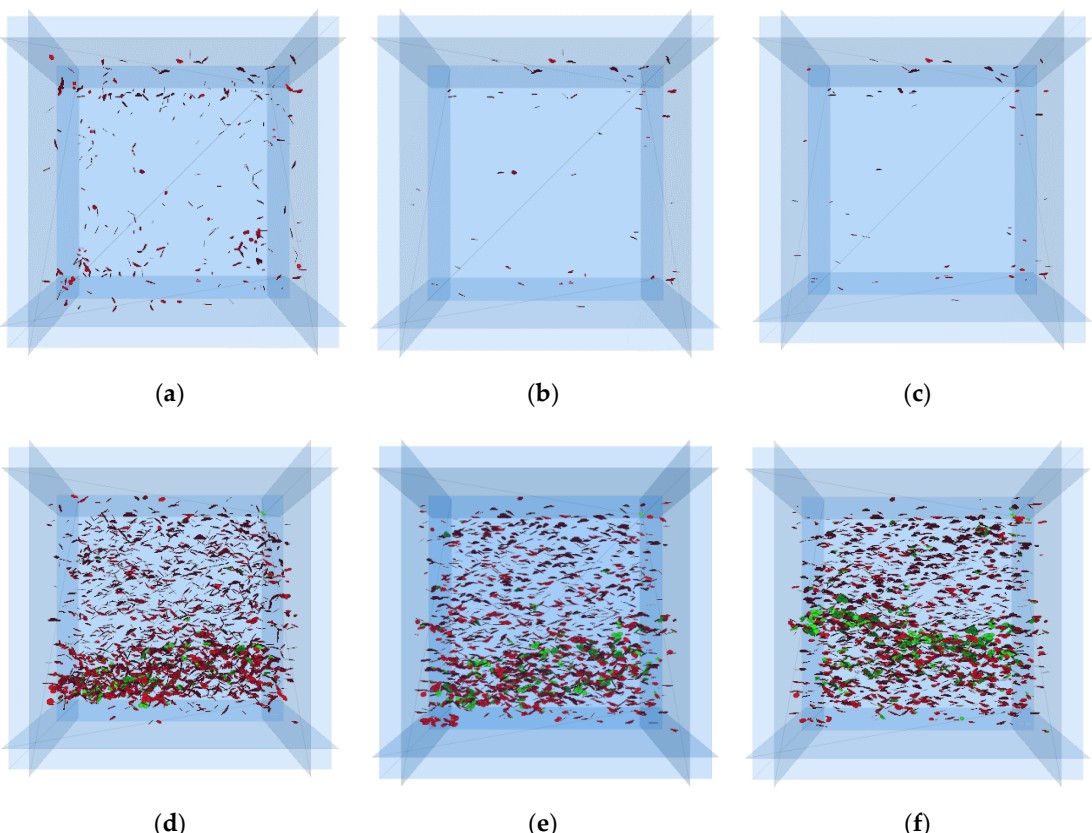

|  |  |  |
|:---:|:---:|:---:|
| (**a**) | (**b**) | (**c**) |
| (**d**) | (**e**) | (**f**) |

**Figure 15.** Comparison of crack formation in the numerical sample for stress path ODEC (Figure 8): (**a**–**c**) after the relaxation of $\sigma_3$ (point E, Figure 8) and (**d**–**f**) at peak strength (point C, Figure 8) for $\sigma_2 = 0$, $\sigma_2 = \sigma_1/2 = 45$ MPa, and $\sigma_2 = \sigma_1 = 90$MPa. Red = tensile crack, green = shear crack. (**a**) $\sigma_1 = 90$ MPa; $\sigma_2 = 0$ (964 cracks); (**b**) $\sigma_1 = 90$ MPa; $\sigma_2 = 45$ MPa (323 cracks); (**c**) $\sigma_1 = \sigma_2 = 90$ MPa (204 cracks); (**d**) $\sigma_1 = 90$ MPa; $\sigma_2 = 0$; (**e**) $\sigma_1 = 90$ MPa; $\sigma_2 = 45$ MPa; (**f**) $\sigma_1 = \sigma_2 = 90$ MPa.

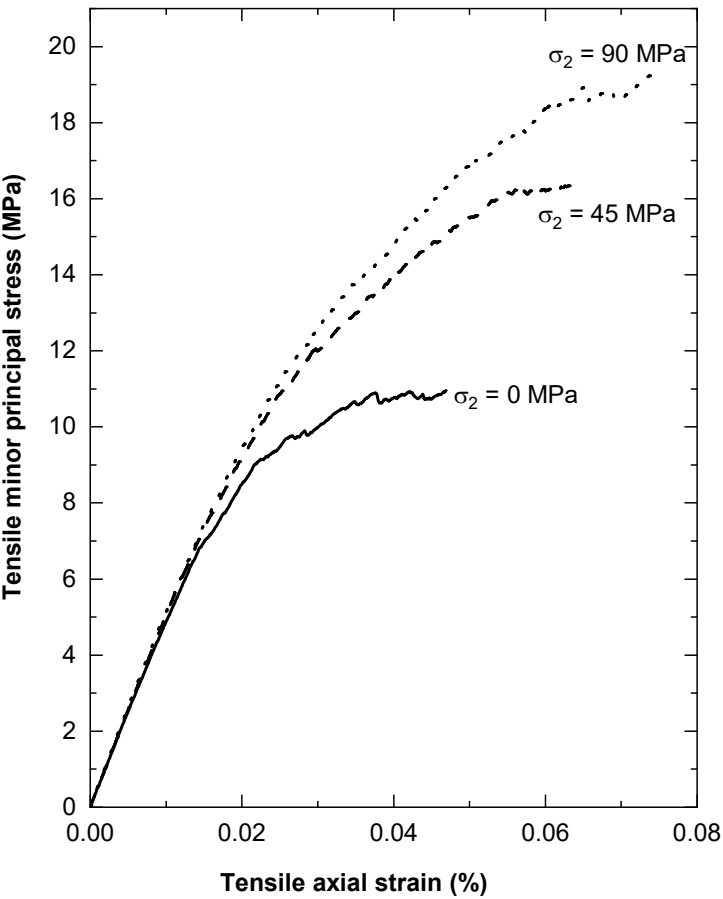

**Figure 16.** Confined extension test at $\sigma_2 = \sigma_1 = 90$ MPa, $\sigma_2 = \sigma_1/2 = 45$ MPa, and $\sigma_2 = 0$ (stress–strain curve showing the impact of $\sigma_2$ in the numerical sample).

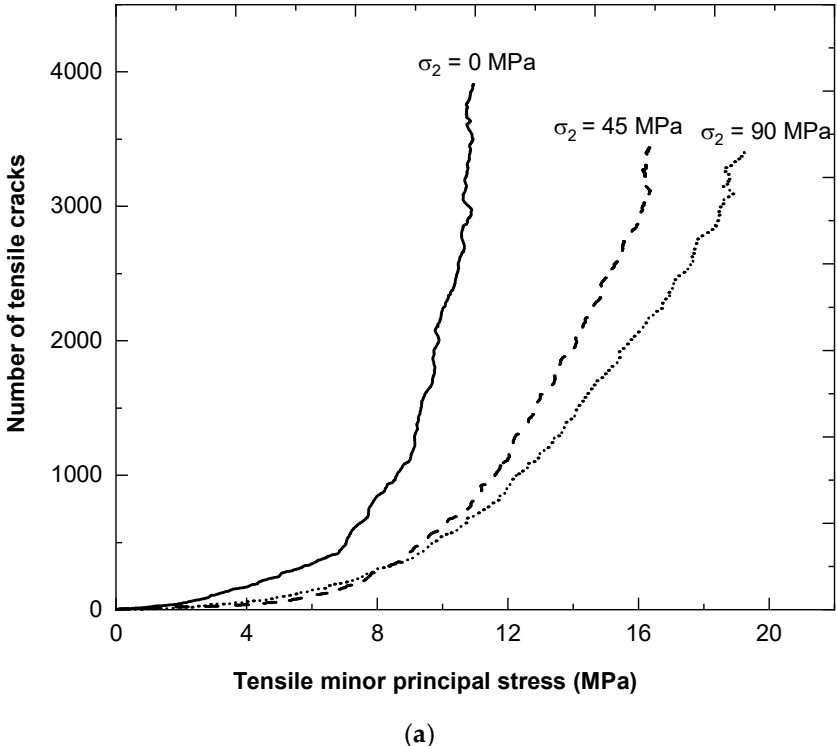

(**a**)

**Figure 17.** *Cont.*

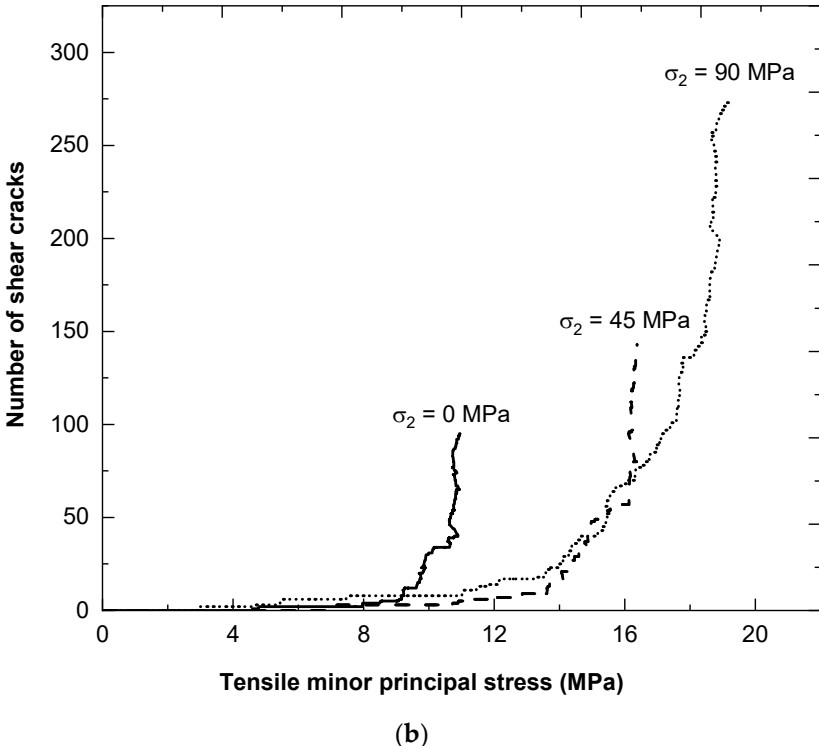

(**b**)

**Figure 17.** Comparison of crack formation in the numerical sample with an increase in $\sigma_2$: (**a**) tensile cracks and (**b**) shear cracks. In all cases, $\sigma_1$ was kept constant at 90 MPa.

## 6. Failure Envelope for Lac du Bonnet Granite

The data points obtained from the direct tension (DT), Brazilian (BR), flattened Brazilian (FB), triaxial, and confined extension tests on numerical samples along with the best fit Hoek–Brown failure envelope obtained from the triaxial tests on Lac du Bonnet granite are plotted in Figure 18. The data points obtained from the confined extension test on the numerical sample pass through the laboratory testing with respect to direct tension, Brazilian and flattened Brazilian, UCS, and triaxial data. The numerical confined extension tests show a tension cutoff up to ~50% of the UCS, after which $\sigma_3$ decreases sharply with the increase in $\sigma_1$. However, as shown in Figure 18, the confinement in the laboratory sample could not be applied beyond 80% of the UCS, which is around $\sigma_{cd}$ for Lac du Bonnet granite. The numerical confined extension test was continued on the compression side ($\sigma_1$ compressive, $\sigma_3 = \sigma_2$ compressive, stress path OFG; Figure 8). Comparing these results to the conventional triaxial test on the numerical sample (stress path OAG; Figure 8) results in a good match and indicates that the influence of the stress path is negligible. When the Hoek–Brown failure envelope was extended into the confined extension region, it was found out that it underestimates actual strength compared to the laboratory direct tension, Brazilian, flattened Brazilian, and confined extension test results on numerical samples. The laboratory direct tension strength was 10.6 MPa, whereas the Hoek–Brown gave a value of 6.8 MPa.

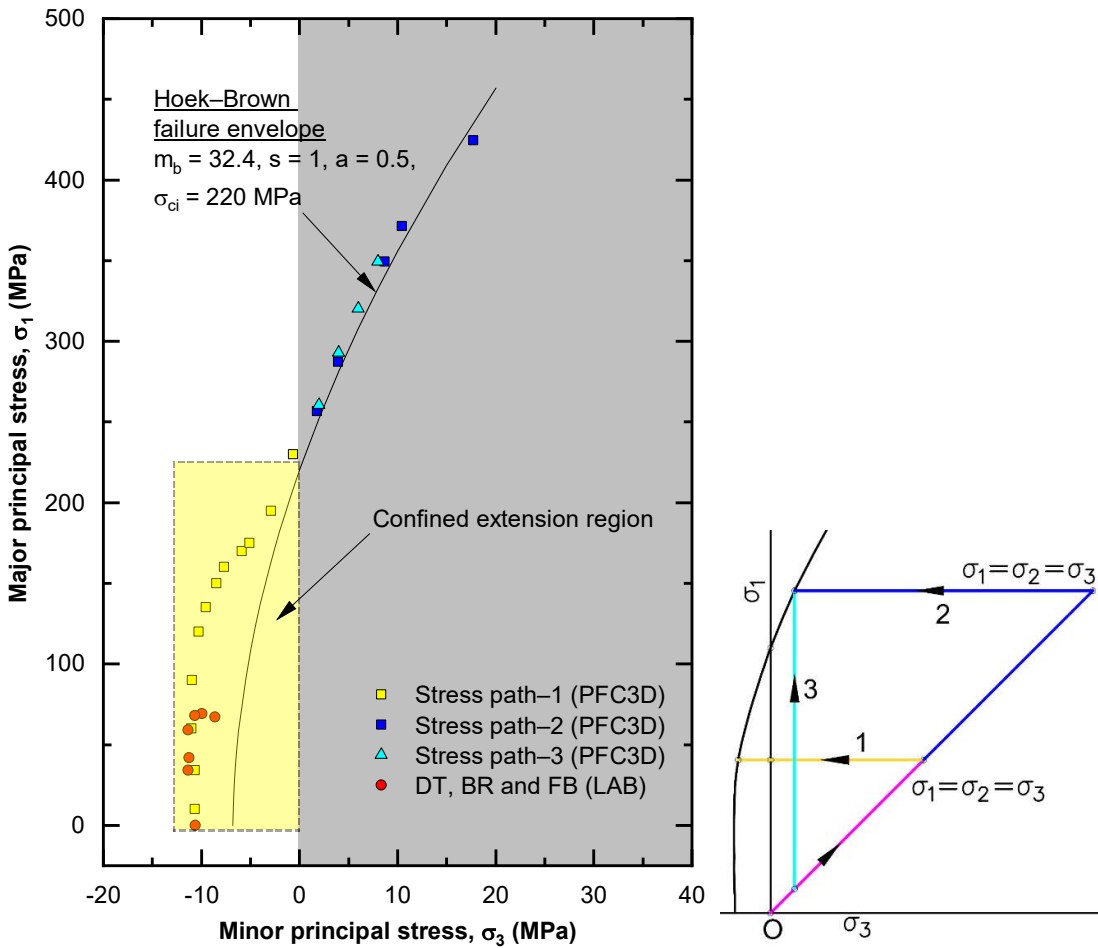

**Figure 18.** Results of laboratory and numerical samples on Lac du Bonnet granite. DT = direct tension, BR = Brazilian, FB = flattened Brazilian.

## 7. Conclusions

The FJ BPM was used in this study to capture the rock behavior observed in the laboratory during UCS, triaxial, and direct tension tests. A methodology was developed to select representative values of the microparameters for the particles and FJs in the numerical sample. With a single mineral grain type with an average of 15 particles along the width of the sample, the model could produce reasonable responses observed for Lac du Bonnet granite in the laboratory during UCS, triaxial, and direct tension tests. The code was modified to study the rock behavior in a confined extension test. The findings can be summarized as follows:

- By changing the code, it was possible to study the behavior of the rock in confined extension at $\sigma_2 = 0$. When compared with the Hoek–Brown failure criterion, the actual strength values obtained from the numerical samples in confined extension were higher than data points from the Hoek–Brown criterion; however, the Hoek–Brown failure envelope overestimated the peak strength obtained from the laboratory tests in confined extension, e.g., as in the case of Carrara marble (Figure 4a).
- The data points obtained from the Brazilian and flattened Brazilian tests were close to the values from the numerical analysis.
- Limited data available from laboratory testing (Brazilian and flattened Brazilian tests) and data from the numerical analysis indicate a tensile cutoff for Lac du Bonnet granite, as suggested by Hoek and Martin [2].

- For the numerical investigation on confined extension, the stress paths investigated have a minor impact on peak strength. However, microfracture formation was found to be path dependent.
- The numerical sample shows a clear impact of $\sigma_2$ in the confined extension test. Rock with 90 MPa confinement for both $\sigma_1$ and $\sigma_2$ produced ~76% higher strength compared to the sample with $\sigma_2 = 0$. This requires a review of the present methodology to test dog-bone-shaped specimens in confined extension.
- Further investigations should consider simultaneous stress rotation along with the stress path to better understand its impact on strength reduction in field conditions compared to the laboratory.

**Author Contributions:** Conceptualization, S.P. and C.D.M.; funding acquisition, C.D.M.; investigation, S.P.; supervision, C.D.M.; writing—original draft, S.P.; writing—review and editing, C.D.M. All authors have read and agreed to the published version of the manuscript.

**Funding:** This research was funded by the Swedish Nuclear Fuel and Waste Management Co. (SKB) Sweden, the Canadian Nuclear Waste Management Organization (NWMO), and the Natural Sciences and Engineering Research Council of Canada.

**Conflicts of Interest:** The authors declare no conflict of interest. The funders had no role in the design of the study; in the collection, analyses, or interpretation of data; in the writing of the manuscript, or in the decision to publish the results.

## Abbreviations

| | |
|---|---|
| $\sigma_{ci}$ | Crack initiation stress |
| $\sigma_{cd}$ | Crack damage stress |
| $\sigma_1, \sigma_2, \sigma_3$ | Major, intermediate, and minor principal stress |
| DT | Direct tension test |
| BT | Brazilian tensile test |
| FB | Flattened Brazilian test |
| UCS | Uniaxial compressive strength |

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
