# Peer review of "Effect of Stress Path on the Failure Envelope of Intact Crystalline Rock at Low Confining Stress"

_minerals, doi:10.3390/min10121119_

Round 1

Reviewer 1 Report

This is an excellent article and most comments are of editorial nature.

Numbers relate to line numbers.

11 Abstract needs to be rewritten. Much is an introduction to paper rather than an abstract of what is in paper. Only from line24 onward is a ‘true’ abstract.

29 UCS/st is not a good keyword

33 poor English to start sentence and even worse paragraph with infinitive.

73, 88 may be worthwhile stating the mean or best-fit failure envelop as data scatter is not shown; 88 explain dashed line

77 … maybe revise to … thereafter, the damage process leading to failure …

79 infinitive again … reverse sentence structure throughout

84 … the other

85 and elsewhere suggest to replace “will be” by “is” …

94 or elsewhere it might be of value to expand to indicate that same damage process occurs during core drilling leading to core damage and possibly reference:

Bahrani N., B. Valley, P.K. Kaiser 2015. Numerical simulation of drilling-induced core damage and its influence on mechanical properties of rocks under unconfined condition. International Journal of Rock Mechanics and Mining Sciences, 80: 40-50.

Article actually does not just focus on unconfined conditions as implied form title, it also deals with stress path related damage during transition to unconfined.

112 … reason for this is yet unknown … ; 41 and 22% seems more appropriate incl in Figure?

115 infinitive

118 FJ not introduce before – I think – and therefore; only in 2.3

120 after Brace

123 mean HB envelop ?

121 Revise Figure labeling: blue dots are not all compression induced tension … if plotted on compression side. You mean compression induced extension … I think.

124 was calculate

133 BPM not introduced … add to 126

145 some readers will argue that an 18 parameter model cannot be calibrated as there i no unique solution. Suggest to add a sentence to the effect … calibration in this context does not mean that only valid combination but one that produces consistent outcomes for various test scenarios … or similar explanation.

158 … use of ‘mimic’ actually enforces this … not a unique set but one that mimics observed behaviours.

160 maybe italic test should be left aligned

172 and elsewhere … maybe replace calibrated by ??? or explain earlier what ‘calibrated’ means.

177, 187 etc … same

212 whereas … and

217 a study does not modify code … reword

237 Results of … FJ BPM in … or ???

279 …the applied BC>.

298 infinitive

310 is this a valid statement of agreement considering that marble is strongly influenced by slip inside minerals?

320 ad DT etc to text for easy relation to figure

321 mean or best fit to data HB

335 clarify … no lab test shown only mean fit

331 ‘confined extension region’ may have to be labelled or shaded to assist reader.

332 reviewer does nto like use of ‘conservative’ depends on use of data … reword underestimates actual strength … or else

334 reword HB gave

340 select representative values

343 also

344 infinitive

346 … the data point or the actual strength … was higher?

349 … did I miss it or was this not modelled … or did model not predict this? Clarify.

355 this conclusion may be mis read … as shown to be higher than back projected HB envelop … maybe revise this bullet

361 clarify … simultaneous stress rotation experienced in the field along the stress path

Reviewer 2 Report

This is a well conducted and presented numerical study on the interrelations between stress path and failure envelope (of crystalline rocks) with some emphasis on the rock mechanical confined extension test and the role of σ2 exemplified for, parameterized with, and compared to experiments conducted on Lac du Bonnet granite.

This study is of both fundamental and applied interest for the rock mechanics community and beyond and certainly appropriate to be published in Minerals.

I have no particular comments and the paper, in my opinion, can be published as is. I only encourage the authors to recheck the abbreviations of technical terms as well as the nomenclature used, e.g., PFC vs. BPM in the abstract; P' vs. P in Section 2.2 and Figure 3.
